# The Spectrum of Thyroid Nodules at Kinshasa University Hospital, Democratic Republic of Congo: A Cross-Sectional Study

**DOI:** 10.3390/ijerph192316203

**Published:** 2022-12-03

**Authors:** John Kakamba Bukasa, Pascal Bayauli-Mwasa, Branly Kilola Mbunga, Ayrton Bangolo, Wivine Kavula, Jean Mukaya, Joseph Bindingija, Jean-René M’Buyamba-Kabangu

**Affiliations:** 1Endocrinology Unit, Department of Internal Medicine, University of Kinshasa Hospital, Faculty of Medicine, University of Kinshasa, Kinshasa, Democratic Republic of the Congo; 2Department of Endocrinology, Liège University Hospital Center, 4000 Liège, Belgium; 3Kinshasa School of Public Health, Faculty of Medicine, University of Kinshasa, Kinshasa, Democratic Republic of the Congo; 4Department of Internal Medicine, Hackensack University Medical Center/Palisades Medical Center, North Bergen, NJ 07047, USA; 5Radiology and Medical Imaging Unit, Department of Internal Medicine, University Hospital of Kinshasa, Faculty of Medicine, University of Kinshasa, Kinshasa, Democratic Republic of the Congo; 6Cardiology Unit, Department of Internal Medicine, University of Kinshasa Hospital, Faculty of Medicine, University of Kinshasa, Kinshasa, Democratic Republic of the Congo

**Keywords:** thyroid nodules, associated factors, Congolese adult

## Abstract

We analyzed the spectrum of thyroid nodules in patients attending the endocrinology unit care of the Kinshasa University Hospital and assessed their associated factors. We conducted a cross-sectional study, performing descriptive statistics and logistic regression. From the 888 enrolled patients, thyroid nodules were detected in 658 patients (74.1%), as mononodules in 22.5% and multiple nodules in 77.5%. Thyroid function was normal in 71.3% cases, while hyperthyroidism and hypothyroidism were found in 26.1% and 2.6% of cases, respectively. Women were more affected than men (75.1% vs. 63.6%; *p* = 0.03). Patients with thyroid nodules were older (44 ± 12 vs. 38 ± 12 years; *p* < 0.001), with a family history of goiter (38.3% vs. 27.4%; *p* = 0.003) and residence in the iodine-deficient region (51.7% vs. 38.8%; *p* = 0.012); they had a higher proportion of longer delays to consultation (47% vs. 20%; *p* < 0.001), but a higher rate of normal thyroid function (85.5% vs. 3 1.3%; *p* < 0.001). Thyroid nodules were associated with the delay to consultation (for duration ≥ three years, OR: 6.560 [95% CI: 3.525–12.208)], multiparity (present vs. absent: 2.863 [1.475–5.557]) and family history of goiter (present vs. absent: 2.086 [95% CI:1.231–3.534]) in female patients alone. The high frequency of thyroid nodules observed requires measures aimed at early detection in the population, the training of doctors involved in the management and the strengthening of technical platforms in our hospitals.

## 1. Introduction

Thyroid nodules are common worldwide, and their prevalence varies depending on whether one considers clinical, ultrasound, or autopsy examination. Thyroid nodules predominate in women and their prevalence increases with age [1,2,3]. The majority of nodules are asymptomatic and benign [4]. The exploration of the nodules involves a clinical examination, completed by an ultrasound using the EU-TIRADS score, the hormonal dosage, the cytopuncture, using the Bethesda score and, of course, the biopsy after the surgery [5,6,7,8,9,10]. Thyroid ultrasound has become a key examination in the management of thyroid nodules [2,10,11,12,13]. In countries with sufficient iodine intake, clinical-based prevalence of thyroid nodules amounts to 5.3–6.4% in women and 0.8–1.6% in men. The prevalence may reach 50% when considering ultrasound, and even 65% when considering autopsy series [1,2,14].

The United States of America is counted among such countries. There is a high prevalence of nodules, with nearly 16 million people with clinically palpable nodules and 219 million with nodules diagnosed on ultrasound [7,15]. This high prevalence is also observed in the European and Asian continents [16,17]. There are a few hospital studies in African settings with variable frequencies depending on the methodology, but there are no randomized studies at national scales to provide prevalence in these countries [18,19]. It should be noted that the prevalence is surging, with a growing number of incidental diagnoses following the development of medical imaging and increased awareness among health professionals [17,20,21,22,23].

Advanced age, iodine deficiency, female sex, obesity, radiation (especially to the cervical region) and a family notion of thyroid diseases are the important risk factors associated with the occurrence of nodules in the general population [24,25,26,27,28]. The literature [5,29] supports that 5 to 10% of these nodules are cancers. Any nodule with lymphadenopathy and rapid growth must be suspected, and a review is required to exclude cancer [30,31]. Despite this overdiagnosis of thyroid cancers, the mortality rate has remained unchanged [32,33,34]. The problem of the management of thyroid nodules is to identify malignant nodules, which must benefit from surgery supplemented by radioactive iodine, radiotherapy or chemotherapy, depending on the histological type found [35,36,37].

In the Democratic Republic of Congo, the management of thyroid nodules encounters many difficulties, namely: the glaring lack of Endocrinologists, radiologists and pathologists (the few specialists available are in the capital of a country whose population is estimated at 100 million inhabitants); poverty and lack of awareness of the population on thyroid pathology; lack of universal health coverage and the insufficiency of technical platforms. All these difficulties explain, on the one hand, the late consultations of specialists by patients and, on the other hand, the impossibility of applying treatment protocols of learned societies. It is important to have data on the extent of thyroid nodules that can be used in advocacy on the need to study the problem at the national level, to encourage universal health coverage, to train specialists in different fields involved in the management of thyroid nodules and to strengthen the technical platforms of our hospitals.

The present study proposes to examine the frequency and the factors associated with thyroid nodules to provide data that can be used in order to justify the study of the problem at the national level, the need to train specialists in Endocrinology and other disciplines involved in the management of thyroid nodules and to strengthen the technical platforms of our hospitals.

## 2. Methods

This clinical study is a cross-sectional study. It was performed in the Endocrinology Department of the University of Kinshasa Hospital. The present study concerned all the patients followed for thyroid pathology during six years from January 2007 to December 2012.

It was an exhaustive sample that enrolled a consecutive series of 888 patients aged at least 18 years old, who attended endocrinology clinics for thyroid pathology and whose medical file contained the results of the thyroid ultrasound, serum levels of thyroid hormones and clinical data. Any patient with a medical file not containing the parameters of interest was excluded.

### 2.1. Variables of Interest

Using a previously established data sheet, investigators sufficiently trained and aware of the objectives of the study collected sociodemographic (age, sex, marital status, province of origin and history of stay in an iodine-deficient region), clinical (family history of goiter, the circumstance of discovery of thyroid pathology, functional and compression signs related to thyroid pathology and gynecological status for women), biological (serum level of thyroid hormones and of thyroid stimulating immunoglobulin) and ultrasound data (measurements of thyroid gland, echostructure and echogenicity of the thyroid gland).

In the present study, thyroid micronodules and macronodules are less than 10 mm and greater than or equal to 10 mm in size, respectively [38,39]. Goiter corresponds to a total volume of the thyroid gland ≥ 18 mL for women and 20 mL for men [40]. Normal thyroid hormone levels were defined as Euthyroidism (T3: 1.2–2.8 nanomoles/L, T4: 65–150 nanomoles/L, or TSH: 0.4–45 mIU/L). A hypothyroidism was defined by TSH > 4.5 µIU/L with T4 < 65 nanomoles/L or T3 < 1.2 nanomoles/L. Hyperthyroidism was defined by TSH < 0.4 mIU/L with T4 > 150 nanomoles/L or T3 < 2.8 nanomoles/L.

### 2.2. Data Analysis

Data were entered into Epidata 3.1 and then transported to SPSS version 25. The results are expressed as mean ± standard deviation (SD) or absolute and relative frequency in per cent. Chi-square and Student t-tests were used for comparing groups with and without thyroid nodules. Multiple regression models and the likelihood ratio method were performed, with thyroid nodule as the dependent variable for the assessment of the strength and independence of association with risk factors in each sex taken separately. A *p*-value ≤ 0.05 was considered statistically significant.

## 3. Results

### 3.1. Sociodemographic and Clinical Characteristics of Patients

The present study enrolled 888 patients aged ≥18 years, of which 811 (91.3%) were females, who attended Endocrinology clinics for thyroid pathology. Their age averaged 42.1 ± 12.1 years; 658 patients (74.1%) had thyroid nodules, 315 (35.5%) reported a family history of goiter and 294 (49.1%) a history of residence in an iodine-deficient region. A thyroid mass (87.8%), heart palpitations (5.2%) and bilateral proptosis (2.4%) were the most frequent of the patients’ complaints. The delay to consultation was < 1 year, 1–2 years and ≥3 years in 24,1%, 35.9% and 40% of patients, respectively. Table 1 indicates a higher frequency of thyroid nodules among women than in men (75.1% vs. 63.6%; *p* = 0.03). Patients with thyroid nodules were older (44 ± 12 years vs. 38 ± 12 years; *p* < 0.001), with a family history of goiter (38.3% vs. 274%; *p* = 0.003) and history of residence in an iodine-deficient region (51.7% vs. 38.8%; *p* = 0.012). The proportions of patients with a normal thyroid function (85.5% vs. 31.3%; *p* < 0.001) and a delay to consultation of ≥3 years (47.3% vs. 20%; *p* < 0.001) predominated among patients with thyroid nodules.

### 3.2. Gynecological Status of Female Patients

As shown in Table 2, the majority of women in the present series were multiparous (70.3%). The frequency of thyroid nodules increased significantly with the parity, from 54.3% in nulliparous to 81.8% in multiparous (*p* < 0.001).

### 3.3. Ultrasound Characteristics of Thyroid Nodules

Among 658 patients with thyroid nodules (Table 3), 148 (22.5%) had one nodule whilst 510 (77.7%) had two or more nodules, the size of which was ≤10 mm in 295 cases (44.8%) or >10 mm in 181 cases (27.5%). In 182 cases (27.7%), mixed-sized nodules (micro- and macronodules) were observed. Solid echostructure (69.9%) and hypoechogenicity (47.1%) were mainly encountered.

### 3.4. Thyroid Nodules Associated Factors

Table 4 shows factors associated to thyroid nodules. In the logistic model, the risk of thyroid nodules significantly (*p* = 0.05 or less) increased with the delay in consultation (for duration ≥ 3 years, OR: 6.560; 95% CI: 3.525–12.208), multiparity (present vs. absent: 2.863 [1.475–5.557]) and family history of goiter (present vs. absent: 2.086 [95% CI:1.231–3.534]) in female patients alone. In men, taken in isolation, no factor emerged as determinant.

## 4. Discussion

To assess the frequency and correlates of thyroid nodules, the present study enrolled 888 patients, with an average age 42.1 ± 12.1 years, of which 811 were females (91.3%), who attended Endocrinology clinics for a thyroid pathology. The saliant results indicate that 74.1% of the series had thyroid nodules that were more frequent among women than men. Patients with thyroid nodules were older, with normal thyroid function in most cases: 22.5% had one whilst 77.7% had two or more nodules. The risk of thyroid nodules increased with the delay to consultation and multiparity among women.

The average age of the patients in the present series agrees with the results of other African studies [26,41,42]. The 658 patients with thyroid nodules were older than those without thyroid nodules. They had one nodule in 22.5% of cases, two nodules or more in 77.7%. Several studies have clearly shown that the number of thyroid nodules do increase with age in the West and Asia [11,25,43,44]. Aging is indeed a risk factor for nodulogenosis. Many morphological and even functional alterations of the thyroid gland do occur with advancing age [25,45,46,47]. The multiplication of nodules with aging can be at least partly accounted for by the delay in seeking appropriate care. The delay to consultation in our patients with thyroid nodules amounted to 3 years or more in almost half of them, which is in keeping with other studies [26,48] that reported that patients with thyroid nodules consulted the doctor late. Furthermore, the thyroid function was unaltered in the majority of cases with thyroid nodules. Such a benign clinical course of the condition, in a setting where universal health coverage is lacking, could also account for the late consultation.

In contrast with our results, a study [49] reported a series with more mononodules (59.5%) than multinodules (40.2%). Such a difference could be explained by the genetic and cultural factors which might vary between African and European populations, and by differences in the access to health care and medical imaging examinations.

Genetic predisposition and environmental features could be invoked to explain the susceptibility to thyroid nodules, as 35.5% and 49.1% of patients with thyroid nodules reported a family history of goiter and a history of residence in iodine-deficient zones. The family notion of goiter is a known risk factor associated with thyroid nodules [33,42,50], especially among twins [26]. The notion of family history of goiter and of residence in given iodine-deficient zones should be capitalized upon to undertake prospective surveys, with the aim to identify potential genetic and environmental causes and their relationships with thyroid nodules in Congolese people.

The predominance of thyroid nodules in female gender is a common observation in the literature [44,51,52,53,54]. It reflects the role of estrogens in the modulation of thyroid structure and function. In our series, multiparity was independently associated with thyroid nodules. During pregnancy, the need of the thyroid hormones increases in order to ensure the normal development, maturation and growth of the fetus. The thyroid gland can enlarge to cover enhanced hormonal needs. Our results concur with the literature, thus concluding that the thyroid nodules are less numerous among nulliparous compared to multiparous [7,15,55,56].

The main patient complaints in our study are similar to those usually reported in the literature [26,35,42]. When present, signs of compression and aesthetic concerns motivate most consultations in Congo. In asymptomatic cases, the discovery of the mass is usually made in by a third party, not by the patient themself. Very often in the African context, people with thyroid masses are marginalized, treated as witches, and suffer from stigma. This appears to constitute a further reason of late consultation.

With regard to the ultrasound data, hypoechogenicity was less frequently observed in the present study (47.1%) in comparison with the series of Zahiri [48], who found 70.8%. The shortcomings of certain ultrasound protocols could account, in part, for this discrepancy.

This study has several limitations, such as missing data or loss of records inherent to its retrospective design, the multiplicity of types of ultrasound devices and operators and the lack of a validated ultrasound examination protocol for the thyroid gland. Furthermore, we were not able to find the indications for the thyroid ultrasound on some of the patients included in our study.

This was a single center observational study. We do not know to what extent the results can be extrapolated to all Congolese patients with thyroid nodules. No risk factors associated with the nodules emerged in men, partially because of the small sample size in the study. However, this study is the first report of data on this topic in our country. It also demonstrates the high frequency of thyroid pathology among Congolese patients.

## 5. Conclusions

The high frequency of thyroid nodules observed requires measures aimed at early detection in the population, strengthening hospital technical platforms and training involved health professionals in improving diagnosis and management.

## Figures and Tables

**Table 1 ijerph-19-16203-t001:** Sociodemographic and clinical features of patients with and without thyroidnodule.

	All Patients *n* = 888	Without Nodules *n* = 230 (25.9%)	With Nodules *n* = 658 (74.1%)	*p*-Value
**Age, years ***	42.1 ± 12.1	37.8 ± 12.2	43.7 ± 11.7	<0.001
**Gender**				0.030
Male	77 (8.7)	28 (12.2)	49 (7.4)	
Female	811 (91.3)	202 (87.8)	609 (92.6)	
**Marital status**				<0.001
Single	271 (31.4)	94 (43.3)	177 (27.4)	
In couple	593 (68.6)	123 (56.7)	470 (72.6)	
**Family history of goiter**				0.003
No	573 (64.5)	167(72.6)	406 (61.7)	
Yes	315 (35.5)	63 (27.4)	252 (38.3)	
**Stay in an iodine-deficient region**				0.012
No	305 (50.9)	74 (61.2)	231 (48.3)	
Yes	294 (49.1)	47 (38.8)	247 (51.7)	
**Delay to consultation, years**				<0.001
<1	212 (24.1)	101 (43.9)	111 (17.1)	
1–2	316 (35.9)	83 (36.1)	233 (35.9)	
3+	351 (40.0)	40 (20.0)	311 (47.3)	
**Complaints**				NA
Thyroid mass	777 (87.5)	151 (65.7)	626 (95.1)	
Palpitation	46 (5.2)	34 (14.8)	12 (1.8)	
Exophthalmia	21 (2.4)	21 (9.1)	0 (0)	
Dysphagia	14 (1.6)	4 (1.7)	10 (1.5)	
Dyspnea	10 (1.1)	6 (2.6)	4 (0.6)	
Weight Loss	20 (2.3)	14 (6.1)	6 (0.9)	
**Thyroid function**				<0.001
Normal	626 (71.3)	72 (31.3)	554 (85.5)	
Hyperthyroidism	229 (26.1)	145 (63.0)	84 (13.0)	
Hypothyroidism	23 (2.6)	13 (5.7)	10 (1.5)	

* expressed as (mean ± SD); NA: Chisquare non-applicable.

**Table 2 ijerph-19-16203-t002:** Gynecological status for all female patients and according to the presence of thyroid nodule.

	All Female Population *n* = 811	Without Nodules *n* = 202	With Nodules *n* = 609	*p*
**Parity**				<0.001
Nulliparous	197 (24.3)	90 (45.7)	107 (54.3)	
Primiparous	44 (5.4)	9 (20.5)	35 (79.5)	
Multiparous	571 (70.3)	104 (18.2)	467 (81.8)	

**Table 3 ijerph-19-16203-t003:** Ultrasound characteristics of thyroid nodules.

	*n =* 658
**Number of Nodules**	
1	148 (22.5)
2 or more	510 (77.5)
**Nodule size**	
Micronodular (≤10 mm)	295 (44.8)
Macronodular (>10 mm)	181 (27.5)
Mixed (Micro and macronodular)	182 (27.7)
**Echostructure**	
Solid	460 (69.9)
Mixed	192 (29.2)
Liquid	6 (0.9)
**Echogenicity**	
Hypoechogenicity	310(47.1)
Hyperechogenicity	112(17.0)
Isoechogenicity	70(10.6)

**Table 4 ijerph-19-16203-t004:** Factors associated to thyroid nodule.

Female				
	Bivariate Analysis		Multivariate Analysis	
	OR_crude_	95% CI	OR _adj_	95% CI
**Age**	1.042	1.028–1.057	1.013	0.987–1.039
**Delay in consultation**				
<1	1			
1–2	2.446	1.652–3.622	2.735	1.561–4.790
3+	4.286	2.418–7.596	6.56	3.525–12.208
**Marital status**				
Single	1			
in couple	2.653	1.861–3.783	0.765	0.399–1.465
**Parity**				
Nulliparity	1			
Primiparity	3.235	1.476–7.089	2.809	0.976–8.080
Multiparity	3.735	2.626–5.313	2.863	1.475–5.557
**History of thyroid disease in family**				
No	1			
Yes	1.655	1.170–2.341	2.086	1.231–3.534
**Stay in goitrogenic zone**				
No	1			
Yes	1.663	1.087–2.545	1.508	0.929–2.448
**Male**				
	**Bivariate analysis**		**Multivariate analysis**	
	**OR_crude_**	**95% CI**	**OR _adj_**	**95% CI**
**Age**	1.044	1.008–1.082	1.002	0.906–1.108
**Delay in consultation**				
<1	1			
1–2	3.106	1.037–9.304	2.232	0.410–12.156
3+	6.538	0.679–62.987	-	-
**Marital status**				
Single	1			
in couple	2.857	1.067–7.653	3.076	0.190–49.803
**History of thyroid disease in family**				
No	1			
Yes	1.655	1.170–2.341	3.943	0.390–39.867
**Stay in goitrogenic zone**				
No	1			
Yes	1.663	1.087–2.545	1.159	0.215–6.250

## Data Availability

Datasets and script files of this research are available as per request to the corresponding author.

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
