# Peer review of "The Spectrum of Thyroid Nodules at Kinshasa University Hospital, Democratic Republic of Congo: A Cross-Sectional Study"

_ijerph, 2022, doi:10.3390/ijerph192316203_

Round 1
Reviewer 1 Report
This paper requires significant improvements in the language, punctuation, and grammar first before this can be properly assessed. I would suggest having someone help the authors with the english and punctuation first before asking reviewers to give their opinion on this article. It is an important study that would add to the literature, however significant improvements to the language etc. is required.
Author Response
Dear reviewer, thanks for taking the time to go over our paper and offer your expert opinion.
Reviewer1
|
(x) Extensive editing of English language and style required |
|
Yes |
Can be improved |
Must be improved |
Not applicable |
This remark was taken into account and language polishing was applied by authors.
|
Does the introduction provide sufficient background and include all relevant references? |
( ) |
( ) |
(x) |
( ) |
The introduction was re-written, and both the content and the presentation were changed. Kindly give us your feedback.

Reviewer 2 Report
The Authors describe the high incidence of thyroid nodules in the Congolese population, referred to a single Hospital. Although this is monocentric cohort with some sample bias, it seems a representative cohort of that Country, where endocrinologists are very rare.
This report is relevant as highlight the need of increasing the awareness of the population and local practitioners about the risk of thyroid nodules
I have some minor comments
1) Did the Authors have any data about thyroid US or incidentally discovered nodules (e.g during supra-aortic trunks doppler scan) in the 416 patients referred for diabetes?
2) The English and the general style require some revisions (eg some capital letters are missing, some sentences should be rephrased etc)
Author Response
Dear reviewer, thanks for taking the time to go over our paper and offer your expert opinion.
Reviewer2
(x) Moderate English changes required
This remark was taken into account and language polishing was applied by authors.
1) Did the Authors have any data about thyroid US or incidentally discovered nodules (e.g during supra-aortic trunks doppler scan) in the 416 patients referred for diabetes?
R/ It is possible that some of nodules included in our study were incidentally discovered. However, we were not able to find the indications for the thyroid US on some of the patients included in our study, and this was added in the limitations of this retrospective study. Lines 212-214.
2) The English and the general style require some revisions (eg some capital letters are missing, some sentences should be rephrased etc)
This remark was taken into account and language polishing was applied by authors.

Reviewer 3 Report
The study describes the characteristics of patients suspected of thyroid disorder, referred to the Kinshasa University Hospital in Kongo. The principal value of the manuscript is that it focuses on an underserved population, which is rarely addressed in medical research.
The thyroid nodules epidemiology in the DRC population has not been assessed, primarily because of the substantial selection bias (patients with neck swelling referred to the tertiary centre). Even the characteristics of the study population are somewhat faltering. The cause of neck swelling in patients negative for thyroid nodules is not given. What were the sequelae of the nodular goitre diagnosis? - fine needle aspiration biopsy? Surgery? And what was the final diagnosis: benign goitre? Thyroid malignancy? The study's main flaw is the lack of a definitive diagnosis while focusing on the nodules' ultrasound characteristics.
The authors stress that the delay in consultation is the risk factor for nodular goitre. It seems that is the reasoning bias, as the earlier talk may be due to rapid neck swelling due to, for example, lymphoma or other diseases with lymph node involvement, making the need for medical consultation more urgent. It seems that the size of the goitre, the size of the dominant nodule (the parameter which has not been addressed) or multinodularity may be affected by the consultation delay (the authors also have not looked into that issue).
There are some minor problems:
- authors have stated that the lack of the goitrogenic substances consumption assessment is the limitation of the study (which is true; however, in the methods section - line 105 - they noted that that issue was addressed.
- the figure lacks the title and seems unfinished
- the results section starts with the sentence left from the instruction for the authors
- the detailed description of the University Hospital structure is unnecessary
- thyroid pathology (thyroid dysfunction? or family history of thyroid diseases?) is mentioned as the risk factor for thyroid nodules (Line 255). The thyroid function is cited as the factor associated with nodularity but not discussed. According to table A1, thyroid dysfunction was less frequent in the nodular goitre patients. Were the patients with thyroid dysfunction consulted earlier than euthyroid ones?
- references are illegible, it requires the reader additional effort to find the cited manuscript
- linguistic editing is necessary.
Author Response
Dear reviewer, thanks for taking the time to go over our paper and offer your expert opinion.
Reviewer 3
(x) Moderate English changes required
This remark was taken into account and language polishing was applied by authors.
The thyroid nodules epidemiology in the DRC population has not been assessed, primarily because of the substantial selection bias (patients with neck swelling referred to the tertiary center). Even the characteristics of the study population are somewhat faltering. The cause of neck swelling in patients negative for thyroid nodules is not given. What were the sequelae of the nodular goitre diagnosis? - fine needle aspiration biopsy? Surgery? And what was the final diagnosis: benign goitre? Thyroid malignancy? The study's main flaw is the lack of a definitive diagnosis while focusing on the nodules' ultrasound characteristics.
R/ Neck swelling referred to thyromegaly. And in our study, out of 888 patients with thyromegaly, only 658 were found to have nodules. The remaining patients had a goiter without nodules. Patients with goiter without nodules served as the control group. They did not undergo FNA as it’s not indicated in patients with goiter without nodules, making it impossible to say if they are malignant or not. Furthermore, surgery was performed in these patients for either esthetic reasons or if they had compressive symptoms.
The authors stress that the delay in consultation is the risk factor for nodular goitre. It seems that is the reasoning bias, as the earlier talk may be due to rapid neck swelling due to, for example, lymphoma or other diseases with lymph node involvement, making the need for medical consultation more urgent. It seems that the size of the goitre, the size of the dominant nodule (the parameter which has not been addressed) or multinodularity may be affected by the consultation delay (the authors also have not looked into that issue).
R/ we agree with the reviewer comment as such the size of the goiter or multinodularity can determine the timing of the consult. In the sense that the bigger the goiter or the more the nodules can lead to an earlier consult. We have to keep in mind that in the population studied on, there is no culture of preventive care as the income of the population is low. So, the population seeks medical attention usually when the symptoms become bothersome.
- authors have stated that the lack of the goitrogenic substances consumption assessment is the limitation of the study (which is true; however, in the methods section - line 105 - they noted that that issue was addressed.
R/ Thank you for this keen observation, that was a mistake, and the issue was addressed.
- the figure lacks the title and seems unfinished
The figure was explaining patient selection. The figure was removed from the paper and the patients selection was explained in the “methods” section. During this study period, 888 patients consulted for thyroid pathologies, of which 658 were found to have nodules and the remaining 230 did not have nodules. The latter group served as the control group. Lines 97-100.
- the results section starts with the sentence left from the instruction for the authors
R/ Thank you for this keen observation, the issue was fixed.
- the detailed description of the University Hospital structure is unnecessary
R/ The description was removed.
- thyroid pathology (thyroid dysfunction? or family history of thyroid diseases?) is mentioned as the risk factor for thyroid nodules (Line 255). The thyroid function is cited as the factor associated with nodularity but not discussed. According to table A1, thyroid dysfunction was less frequent in the nodular goitre patients. Were the patients with thyroid dysfunction consulted earlier than euthyroid ones?
R/ Family history was cited as risk factor for thyroid nodules; however, thyroid dysfunction is not cited in our paper as a risk factor for thyroid nodules. This can be found on table 4. And yes, thyroid dysfunction was less frequent in the nodular goiter patients. The data made available to us does not contain information on the timing of consultation for euthyroid patients and those with thyroid dysfunction.
- references are illegible, it requires the reader additional effort to find the cited manuscript
R/ Thank you for bringing this matter to our attention. We improved the quality of our references.
